# Global Meta-path-level Counterfactual Explanation for Heterogeneous Graph Neural Networks by Path Exclusion

## Abstract

Heterogeneous graph neural networks (HGNNs) capture rich structural and semantic information in real-world data, but their decision processes are often opaque. While recent work in graph counterfactual explanations perturbs nodes, edges, or subgraphs to identify influential components, such approaches are too coarse for heterogeneous graphs: a single perturbation can disrupt many meta paths, obscuring which relations truly drive model behavior. Since meta paths are fundamental units of semantic information in HGNNs, explanations at the meta-path level are both natural and necessary. We introduce meta path exclusion, a framework for global counterfactual explanations that directly perturbs specific meta paths. To achieve this, we propose the path exclusion algorithm, which modifies the forward pass of HGNNs to exclude target meta paths while preserving the rest of the graph, enabling precise attribution of model performance to individual paths. Experiments on four benchmark datasets (DBLP, OGB-MAG, IMDB, and MovieLens) show that excluding a small set of meta paths can significantly reduce accuracy, revealing their critical role. Moreover, when these identified paths are reused in meta-path-dependent HGNNs, their removal consistently degrades accuracy by 5–12%, confirming their general importance. Our results establish meta paths as atomic units for fine-grained, faithful counterfactual explanations in heterogeneous graphs.

## 1 Introduction

Graph neural networks (GNNs) have gained significant recognition for their effectiveness in processing graph-structured data, excelling in tasks such as node and edge classification (Li et al., 2025; Zhao et al., 2024a; Ding et al., 2024; Cheng et al., 2024), link prediction (Mattos et al., 2024; Zhao et al., 2024b; Cho, 2024), and graph classification (Yao et al., 2024; Liu et al., 2024). Although GNNs excel in such classification tasks, their deployment in real-world scenarios faces a critical challenge: many real-world graphs are heterogeneous, containing multiple node and edge types with distinct semantics. In such cases, researchers typically specify a meta-structure-—-a predefined schema that enumerates the node/edge types and the permissible relations among them-—-so that GNNs can reason over relation-specific patterns rather than treating all connections uniformly (Figure 1a).

Within this schema, an important concept is the **meta path**, which represents a sequence of node and edge types. Meta paths capture higher-order semantic relations that go beyond direct connections. For example, in an academic network, the meta path *Author-Paper-Venue-Paper-Author* connects two authors if they publish in the same venue, reflecting an indirect collaboration. Such paths have been widely used to encode domain knowledge and enhance the performance of heterogeneous GNNs (HGNNs) (Dong et al., 2017; Wang et al., 2019; Fu et al., 2020), as they explicitly model real-world relationships through a predefined set of meta paths. However, the performance of HGNNs is highly sensitive to which meta paths are selected.

To avoid the burden of manually selecting paths, meta-path-independent HGNNs have been proposed (Chen et al., 2019; Busbridge et al., 2019; Park et al., 2022; Yang et al., 2023b; Zhang et al., 2019; Hu et al., 2020b; Jin et al., 2021). These models automatically learn weights over all pos-

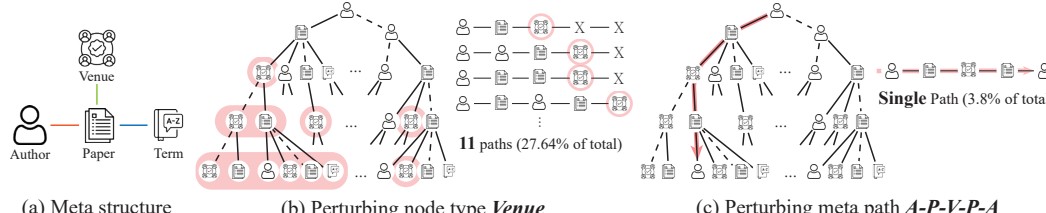

(a) Meta structure      (b) Perturbing node type **Venue**      (c) Perturbing meta path **A-P-V-P-A**

Figure 1: The DBLP dataset example, illustrating the impact of excluding both nodes and meta path within the meta structure of a heterogeneous graph.

sible paths in the meta structure. This increases flexibility and often improves performance, since the model can discover useful paths overlooked by human experts. However, it also obscures transparency: once the model learns implicit weights over potentially thousands of paths, it becomes unclear which paths are truly responsible for the prediction. As a result, practitioners cannot easily verify whether the model is relying on meaningful semantic relations (e.g., co-authorship through a venue) or on spurious shortcuts. This lack of interpretability limits the trustworthiness and practical deployment of such models.

This raises the challenge of explainability in HGNNs. In general, explainable GNNs aim to identify the graph elements most responsible for a prediction. One popular approach is the counterfactual explanation: finding minimal changes to the input that alter the model output (Amitai et al., 2024; Wang et al., 2023b; Estornell et al., 2023). In homogeneous graphs, counterfactuals can be generated by perturbing nodes (Ying et al., 2019), edges (Vu & Thai, 2020), or subgraphs (Yuan et al., 2021), but for heterogeneous graphs, these perturbations are often too coarse. Consider again the academic network. The prediction task often involves classifying the research field of an author. One useful signal is that authors working in the same field tend to publish in similar venues. The meta path *Author–Paper–Venue–Paper–Author* captures this relation, connecting authors who publish in the same venue. In the legacy way, we can perturb a node type *Venue* to measure how the node type is important in the given HGNN. **However, this approach introduces a 'ripple effect.'** For instance, perturbing a single 'Venue' node inadvertently disrupts every meta path involving that venue as shown in Figure 1b. Consequently, it becomes difficult to distinguish whether the model's prediction relies on a specific semantic relation (e.g., co-authorship) or merely the presence of the node itself. What is truly needed is a counterfactual explanation at the meta-path level. For example: "the model predicted these two authors are similar because they both published in the same venue; if we remove that meta path, the prediction changes."

To address this gap, we propose *global meta-path-level counterfactual explanations*. Instead of perturbing nodes or edges, we directly perturb specific meta paths, which serve as finer-grained and semantically meaningful units. We introduce *path exclusion algorithm*, a post-hoc method that excludes target meta paths during the forward pass of a trained HGNN. This allows us to evaluate the influence of each meta path without distorting the broader graph structure.

Our experiments on three heterogeneous node classification and a edge classification datasets show that removing a small fraction of path instances can lead to significant performance drops (Section 5.2). For instance, excluding the *Author–Paper–Venue–Paper–Author* meta path in DBLP, which accounts for only 3.8% of path instances, changes the model's prediction by 44.4%. In comparison, removing the entire *Venue* node type—accounting 27.6% of paths—produces a similar accuracy drop. These results demonstrate that meta-path-level perturbations provide more precise and interpretable explanations than traditional approaches. Furthermore, we validate the utility of our explanations by reusing identified critical meta paths in meta-path-dependent models (Section 5.3). Removing these high-impact paths consistently lowers accuracy by 5–12%, confirming their importance across architectures.

In summary, our contributions are listed below: (The code for our experiments is publicly available[1].)

- We uncover the need for comprehensive explanations at the meta-path-level and introduce the concept of global meta path counterfactual explanation for HGNNs.

---

[1]https://doi.org/10.5281/zenodo.11218749

- We propose *path exclusion algorithm*, which is a post-hoc algorithm designed specifically for excluding meta paths from a HGNN. The algorithm allows us to perturb meta paths in HGNNs that were previously inaccessible.
- We conduct experiments to evaluate the efficacy of fine-grained explanation through meta path exclusion and demonstrate the application to identify globally critical meta paths by leveraging meta-path-dependent models.

## 2 RELATED WORKS

**Explanations of Graph Neural Networks** Research on explaining graph neural networks has explored various levels of granularity in graph components, such as nodes, edges, and subgraphs. While a few studies have addressed global explanation for GNN (Xuanyuan et al., 2023; Azzolin et al., 2022), the majority have focused on local explanation, elucidating the importance of graph components during model predictions on specific input samples (Ying et al., 2019; Vu & Thai, 2020; Xiong et al., 2023). GNNExplainer (Ying et al., 2019) was the pioneering approach to explaining graph neural networks, tailoring a method specifically for GNNs by masking node features to maximize mutual information with subgraphs. Following a similar approach, PGExplainer (Luo et al., 2020) parameterized the optimizer as a multilayer perceptron and formulated the problem as a graph structure, focusing on the edge-level explanation. PGMExplainer (Vu & Thai, 2020) employed a surrogate model explanation method using probabilistic graphical models to explain GNNs at the node-feature-level. GNN-LRP (Xiong et al., 2023) explains GNNs at the walk-level, proposing a layer-wise relevance propagation aligned with walks.

**Counterfactual Explanations for GNN** Counterfactual explanation has emerged as a powerful method for causally comprehending models in post-hoc settings. Efforts have been made to counterfactually explain graph neural network models (Numeroso & Bacciu, 2021; Bajaj et al., 2021; Lucic et al., 2022; Ma et al., 2022). MEG (Numeroso & Bacciu, 2021) was the first work on graph counterfactual explanation, employing a heuristic algorithm to find a counterfactual subset of edges for molecular graphs. Following this approach, Bajaj et al. (2021) proposed robust counterfactual algorithms for GNNs by analyzing graph data into subsets of edges leveraging linear decision boundaries. CF-GNNExplainer(Lucic et al., 2022) decomposes graph data into node representation vectors, generating counterfactual examples that take subsets of representation vectors, resulting in a binary perturbation matrix for nodes in the graph. CLEAR (Ma et al., 2022) proposes generative graph counterfactual explanation, resulting in a counterfactual stochastic graph similar to the original one using a variational auto encoder. Specific for the heterogeneous graph, CF-HGExplainer (Yang et al., 2023a) tries to locally explain HGNN by perturbing node and edges, and xPath (Li et al., 2023) also propose a local counterfactual explanation in path-level considering path addition.

More recently, there have been a few works on global counterfactual explanation with limited recourse conditions (Kosan et al., 2022; Huang et al., 2023; Wang et al., 2023a). Kosan et al. (2022) pioneered this field, narrowing the recourse to Outer-If/Inner-If/Then conditions for a global counterfactual explanation method targeting general models. For GNN explanation applications, Huang et al. (2023) utilized the edit map of the original graph for GNN global counterfactual explanation. Wang et al. (2023a) proposed another global graph counterfactual explanation method for molecule classifier networks, leveraging the idea of a global common-neighbor-graph-as-explanation.

Unlike the past literature, this paper proposes a meta path exclusion algorithm for global counterfactual. While other research focuses on optimization techniques to identify nodes or edges for perturbation, we take a step back to novel question on the fundamental unit of perturbation itself.

## 3 PRELIMINARIES

### 3.1 META PATHS IN HETEROGENEOUS GRAPH DATA

Our primary focus for explanation lies within the meta structure of heterogeneous graphs. Unlike homogeneous graphs, which consist of nodes and edges without distinct types, nodes and edges in a heterogeneous graph are pre-assigned specific types. We refer to the graph structure of these predefined types and their relationship as a meta structure of a heterogeneous graph; the sequence of connected edge types, rather than an actual instance of the path, is a meta path. For example,

consider an academic graph data where an author named *A* authored the paper titled *XAI* in the venue *ICLR*. The virtual sequence of *Author-Paper-Venue* represents a meta path, while the actual sequence *A-XAI-ICLR* represents an instance path of this meta path. Below are the formal definitions of a heterogeneous graph and a meta path.

**Definition 3.1 (Heterogeneous Graph and Meta Structure)** *A heterogeneous graph $\mathcal{G}$ is defined as a directed graph $\mathcal{G} = (\mathcal{V}, \mathcal{E})$, where there exists a node type mapping $\phi : \mathcal{V} \rightarrow \mathcal{A}$ and an edge type mapping $\psi : \mathcal{E} \rightarrow \mathcal{R}$. Here, $\mathcal{A}$ and $\mathcal{R}$ denote the sets of node types and edge types, respectively. A graph composed of node types and edge types, $\mathcal{G}_M = (\mathcal{A}, \mathcal{R})$, is a meta structure of the original heterogeneous graph $\mathcal{G}$.*

**Definition 3.2 (Meta path)** *Given a heterogeneous graph $\mathcal{G} = (\mathcal{V}, \mathcal{E})$, a finite meta path $P$ is defined as a sequence of node types $(A_1, A_2, ..., A_n)$, where $A_i \in \mathcal{A}$ for $0 < i \leq n$, satisfying that $R_i$ represents an edge type between node types $A_i$ and $A_{i+1}$. For a meta path $P$, a meta path instance $p = (e_1, e_2, ..., e_{n-1})$ is a path in $\mathcal{G}$ that satisfies $\psi(e_i) = R_i$.*

## 3.2 PROBLEM DEFINITION

In this study, we introduce the novel problem of *global counterfactual explanation by the meta path* for meta structure of heterogeneous graph neural network (HGNN) models. Here, the modification on a meta structure is limited to deletion because addition or substitution cannot be well-defined.

Before defining the global counterfactual problem, we first introduce its local counterpart by adapting existing graph counterfactual approaches (Lucic et al., 2022). Specifically, we modify the input perturbation process to focus on meta path exclusion rather than traditional node or edge modifications. Accordingly, we define the concept of *local meta path counterfactual explanation for HGNN model* as follows:

**Problem 3.1 (Local Meta Path Counterfactual Explanation)** *Given a trained HGNN model $\Phi$ and an input heterogeneous ego graph $G_i(V, E)$ for a node $i$, a set of meta paths $P$ in $\mathcal{G}_M$ constitutes a local meta path counterfactual explanation of $\Phi$ for node $i$, where $\Phi(G_i) \neq \Phi(G_i/P)$. Here, $\Phi(G_i/P)$ is the model prediction of $\Phi$ where only meta paths $P$ are ignored from calculation of $\Phi(G_i)$.*

To extend a local method into a global one, the counterfactual explanation necessitates recourse, as stated in Kosan et al. (2022); Huang et al. (2023), to filter the input graph instances based on whether they are covered by a local explanation or not. Existing global counterfactual explanation approaches for graphs employ the edit map of graph components (such as nodes, edges, subgraphs) as the recourse. For HGNN, the meta structure of the heterogeneous graph data is a predefined common scheme over whole dataset, so the modified meta structure serves as a natural recourse because it is shared by every data instance in a dataset. Hence, the recourse quality of the global counterfactual explanation is defined as $N(\{G \in \mathbf{G} \mid \Phi(G) \neq \Phi(G/P)\})$, which denotes the number of data points that changes the final label by excluding the meta path set P. Formally, the problem of finding effective global meta-path-level explanation is defined as below:

**Problem 3.2 (Global Meta-path-level Counterfactual Explanation)** *Given a HGNN model $\Phi$ and a heterogeneous graph $\mathcal{G} = (\mathcal{V}, \mathcal{E})$, global meta-path-level counterfactual explanation problem entails optimizing a set of meta paths $P = \{p \mid p$ is a meta path, $N(p) = k\}$ as $max_P$ $N(\{G \in \mathcal{G} \mid \Phi(G) \neq \Phi(G/P)\})$, where $\mathcal{G} \simeq \mathcal{G}/P$.*

In this problem, we leave the possibility open for the definition of similarity $\mathcal{G} \simeq \mathcal{G}/P$ for the use of later research. The definition we used in this work is described in Section D.1.

## 4 METHOD

### 4.1 OVERVIEW

The process of global meta-path-level counterfactual explanation consists of two key stages, similar to general counterfactual explanations: perturbation and optimization. In the perturbation stage,

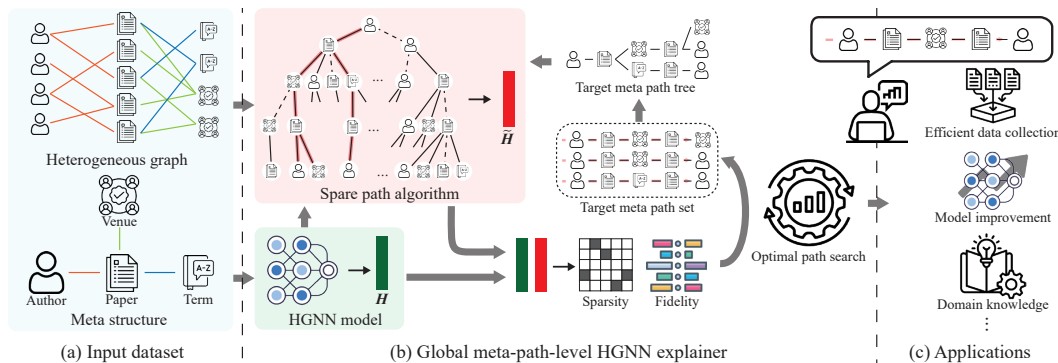

Figure 2: Overall workflow of global meta-path-level HGNN explanation.

given a heterogeneous graph and a trained HGNN model, node embeddings are computed while excluding the target meta path set. The following section (Section 4.2) provide an in-depth discussion of the challenges and solutions associated with meta path exclusion (i.e., perturbation) in HGNN explanation. In the optimization stage, the perturbed embeddings are evaluated (details in Section D.1), and the optimal meta path set is identified. As our primary goal is to demonstrate the effectiveness of meta path perturbation in HGNN explanation, we do not propose a novel optimization method but instead apply existing optimization techniques alongside our meta path exclusion approach (details in Appendix E.5). The identified critical meta path set in HGNNs has multiple applications, including improving data efficiency, enhancing model performance, and extracting domain knowledge. The practical applicability of this approach is demonstrated in Section 5.3, where we assess model accuracy by varying the predefined meta path set used for training meta-path-dependent HGNNs. Figure 2 illustrates the overall workflow of global meta-path-level HGNN explanation.

## 4.2 PROPOSED METHOD

### 4.2.1 SINGLE PATH EXCLUSION

The conventional approach to counterfactual explanation in GNN models involves the removal of the component of interest from an input data point without changing the model inference step (Lucic et al., 2022; Numeroso & Bacciu, 2021). Given a partition of a graph as a batch, the counterfactual methodology omits a node and its surrounding edges for node exclusion, an edge for edge exclusion, or a sub-graph for sub-graph exclusion. These methods alter the adjacency matrix, one of the input of GNN models. However, paths pose a challenge as they cannot be directly modified from the input data by controlling the adjacency matrix. When any tangible component of a graph is removed, multiple paths can concurrently be excluded, as depicted in Figure 1. Consequently, it is necessary to design an algorithm that adjusts the entire forward inference of the model rather than perturbing the input data directly. We propose a wrapper algorithm for the forward function, called the *path exclusion algorithm*, which adjusts the computation of the input data during the calculation of each layer (outlined in Figure 2).

Given that the computation graph of an HGNN is in a tree structure, perturbing the path in the computation graph can be achieved by removing a specific leaf in the first layer, effectively eliminating the corresponding path. Formally, while the normal first convolution layer of the HGNN is calculated as

$$H_1 = Conv(H_0, \mathcal{E}) \tag{1}$$

eliminating the specific leaf on the meta structure modifies the equation to

$$\tilde{H}_1 = Conv(H_0, \mathcal{E}_{CF}), \tag{2}$$

where $\mathcal{E}$ denotes the set of edges, and $\mathcal{E}_{CF} = \{e | e \in \mathcal{E} \text{ and } \psi(e) \neq R\}$ is the set of edges excluding the edge type $R$, connecting the leaf nodes corresponding to the target meta path. Here, $H$ stands for the normal embedding while $\tilde{H}$ denotes the modified embedding.

In GNN, node embeddings are computed once per layer. Therefore, removing a leaf node in the first layer affects the embeddings of all subsequent layers for nodes that are ancestors of the removed leaf in the first layer.

For example, in Figure 3, when a meta structure is formed in a triangular shape with node types A, B, and C, and given the HGNN model composed of three layers, the calculation of the embedding of node type A includes two groups of meta paths containing node type A in the third entity: *A-B-A-X* and *A-C-A-X*. Due to the current implementation of HGNN, the embedding of A in the second HGNN layer is computed by the same value in both paths, but when excluding meta path *A-C-A-B*, those embeddings must have different values. For the case where the target meta path is *A-B-A-X*, the embedding A in the third layer aggregates embedding B and C normally, while the embedding A' from *A-C-A-X* aggregates only C, excluding B. Thus, to exclude the path from the computation graph, we should modify the implementation to store both the normal embedding $H_k$ and the modified embedding $\tilde{H}_k$.

To address this, we introduce the concept of *spare embeddings*. The core idea is to maintain two sets of computation graphs: one for the standard forward pass and the other for the path-excluded version. By caching the original (normal) embeddings, we can selectively combine them with the perturbed embeddings only where necessary. To calculate embeddings where meta paths are excluded from the computation, both the normal embedding $H$ and the embedding without specific leaves $\tilde{H}$ are computed in the first layer using the equations above. The modified embedding is *spared* for use in calculating the node embedding relevant to the target path. From the next layer, the spare embeddings are calculated by combining spare embeddings and normal embeddings, while the normal embeddings are calculated as usual in HGNN. When calculating the spare embedding, the input embeddings of the neighboring nodes are the spare embeddings from the previous layer only if the edge type connecting the target node and neighbor corresponds to the excluding meta path. Otherwise, the neighboring nodes in the spare embedding use the normal embedding from the previous layer. Formally,

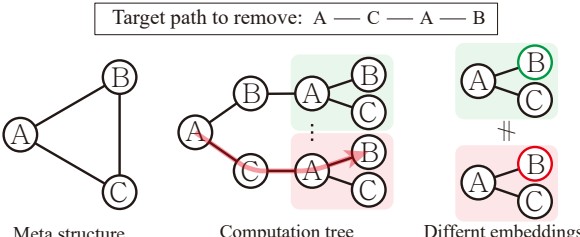

Figure 3: Illustration of the example of a triangular meta structure. In the perturbed computation, two different values need to be input into the same node type B to calculate A in different paths.

$$\tilde{h}_k^i = \begin{cases} Conv(\tilde{H}_{k-1}, \mathcal{E})^i & \text{if } (k, \phi(i)) \in R_{CF} \\ Conv(H_{k-1}, \mathcal{E})^i & \text{otherwise} \end{cases} \tag{3}$$

where $R_{CF}$ is the set of $(k, A_x)$ such that the target path contains the node type $A_x$ in the $k$th layer. Finally, the spare embedding of the final layer becomes the embedding without the target path.

### 4.2.2 MULTIPLE PATH EXCLUSION

The basic logic behind multiple path exclusion is similar to the single path exclusion method. Our algorithm (Algorithm 1) exclude the computation for the leaf part of the excluding path in the first layer of the HGNN. The combination of the spare embedding, the leaf-excluded embedding, and the normal embedding results in the final embedding without the specific meta path.

However, for multiple path exclusions, spare embeddings should be computed for each target meta path, and combining these embeddings over the meta structure is not straightforward. Extending the single path algorithm to multiple paths requires handling dependencies between shared ancestors. When multiple target paths diverge from a common node in the computation tree, their exclusion effects must be aggregated.

For example, when excluding *A-B-A-B* and *A-B-C-A* from the triangular graph in Figure 3, the second GNN layer should calculate B's spare embedding from both spare embeddings A (from the *A-B-A-B* path) and C (from the *A-B-C-A* path). Conversely, the first GNN layer should separately calculate A's spare embedding for each path because they do not share ancestors.

We manage this by constructing a target meta-path tree ($T_P$) of target meta paths ($P$), which tracks specific branches for exclusion while preserving the shared structure. Formally,

$$\tilde{h}_{k,n}^i = \begin{cases} Conv(\tilde{H}_{k-1,m}, \mathcal{E})^i & \text{if } m \in \mathcal{C}(n) \text{ and } \mathcal{A}(i) = t(m) \\ Conv(H_{k-1}, \mathcal{E})^i & \text{otherwise} \end{cases} \tag{4}$$

where $\mathcal{C}(n)$ is the set of child nodes in the target path tree, and n and m are the nodes in the tree. Also, $t(m)$ represents the node type of node m in the tree. Finally, the spare embedding corresponding to the (final layer, root node) becomes the embedding without the target meta paths.

# 5 EXPERIMENT

## 5.1 EXPERIMENTAL SETTINGS

We provide a summary of our experimental setup below, with detailed descriptions in Appendix D.

- **Metrics.** To evaluate counterfactual explanations, we use three complementary metrics: *Fidelity*, the proportion of predictions that change under the counterfactual; *Distance*, the number of meta paths perturbed; and *Sparsity*, the fraction of meta path instances affected by the perturbation.

- **Target Model and Baselines.** We adopt the Heterogeneous Graph Transformer (HGT) (Hu et al., 2020b) as the target model. For comparison, we adapt baselines to the heterogeneous setting including *CF-GNNExplainer* (Lucic et al., 2022), *PGM-Explainer* (Vu & Thai, 2020), and *Attention*-based explanation (Hu et al., 2020b).

- **Datasets.** Experiments are conducted on three widely used heterogeneous graph benchmarks: *DBLP* (Fu et al., 2020), *OGB-MAG* (Hu et al., 2020a), *IMDB* (Fu et al., 2020), and *MovieLens*[1].

- **Implementation Details.** For the target model, we selected the heterogeneous graph transformer (HGT) (Hu et al., 2020b) due to its state-of-the-art performance across diverse tasks and datasets. The model was trained on three node classification datasets and a edge classification dataset with four HGT layers of embedding dimension 64-256 (the best model is selected for each dataset) and a fully connected classification layer. The depth is set to four because the original meta-path-dependent models consist of paths with a maximum of four edges. We find the global minimum of fidelity under constraints on sparsity and distance by brute-force searching for following experiments, as our **main focus is to demonstrate the effectiveness of the meta-path-level explanation rather than optimization**.

## 5.2 EXPLANATION RESULTS

### 5.2.1 EXPLANATION WITH SINGLE PATH EXCLUSION

Table 1 presents the results of single path explanations with the top fidelity. The findings underscore the effectiveness of meta-path-level explanations over perturbing other graph components, particularly in achieving low fidelity with high sparsity, compared to perturbations on coarser graph components. In the DBLP dataset, the meta-path-level explanation demonstrates significant fidelity (55.57%), surpassing half of the base model accuracy (86.74%), with considerably higher sparsity compared to any coarser approach yielding similar accuracy degradation. Specifically, the exclusion of meta path *A–P–V–P–A*, which has the lowest fidelity, displays a sparsity score of 96.20%, while perturbing other graph components resulting in fidelity below 60% show sparsity under 91%. In the OGB-MAG dataset, although the fidelity is lower than other coarser explanations, the sparsity is substantially higher (over 99.4%) than the others. For the IMDB dataset, the maximum fidelity is 98.54%, while the meta path with the highest sparsity (99.54%) exhibits accuracy degradation.

Another noteworthy aspect of the results is the latent connection between path explanation and other components. In the DBLP dataset, the path *A–P–V–P–A* shows overwhelming fidelity, while in

---

[1]https://movielens.org/

Table 1: Results of counterfactual explanation. Each group of rows presents the top-5 meta-structure components identified as most important by each explanation method. For methods operating at the edge or node level, explanations are reported in the format (layer, edge/node type). All values are percentages (%). Dataset-header accuracy refers to the base classifier accuracy.

| Method (Granularity) | DBLP (Accuracy 86.74%) | | | OGB-MAG (Accuracy 40.36%) | | | IMDB (Accuracy 58.05%) | | | MovieLens (Accuracy 32.95%) | | |
|---|---|---|---|---|---|---|---|---|---|---|---|---|
| | Expl. | Fidelity↓ | Sparsity↑ | Expl. | Fidelity↓ | Sparsity↑ | Expl. | Fidelity↓ | Sparsity↑ | Expl. | Fidelity↓ | Sparsity↑ |
| Ours (Meta Path) | APVPA | 55.57 | 96.20 | PAPAI | 97.07 | 99.63 | MDMDM | 98.54 | 99.54 | UMUMU | 35.19 | 99.88 |
| | APVPT | 93.80 | 91.68 | PPPPF | 97.52 | 99.93 | MDMAM | 99.942 | 90.89 | MUMUM | 65.80 | 68.78 |
| | APVPV | 95.49 | 99.99 | PPPAI | 98.71 | 99.87 | MAMDM | 99.885 | 90.89 | | | |
| | APTPV | 97.11 | 99.94 | PPAPF | 98.80 | 99.41 | MAMAM | 99.972 | 67.74 | | | |
| | APAPA | 98.16 | 99.97 | PAPPF | 98.94 | 99.40 | | | | | | |
| CF-GNNExplainer (Edge) Lucic et al. (2022) | 4, A–P | 41.54 | 18.36 | 1, P–F | 82.8 | 82.57 | 1,D–M | 93.71 | 89.01 | 1, M–U | 5.96 | 93.70 |
| | 1, P–A | 49.03 | 90.77 | 4, P–A | 91.13 | 82.19 | 1,M–D | 94.23 | 93.59 | 3, M–U | 6.23 | 76.52 |
| | 2, V–P | 52.90 | 80.11 | 2, A–P | 92.26 | 94.12 | 4,M–D | 94.69 | 84.80 | 2, U–M | 6.55 | 76.52 |
| | 3, P–V | 53.30 | 72.36 | 1, A–I | 93.73 | 97.45 | 2,M–D | 95.23 | 97.59 | 4, U–M | 6.58 | 93.70 |
| | 1, P–T | 84.53 | 81.87 | 3, A–P | 94.02 | 87.43 | 3,D–M | 96.07 | 84.80 | 4, M–U | 25.18 | 36.82 |
| PGM-Explainer (Node) Vu & Thai (2020) | 4, Paper | 40.65 | 18.36 | 2, Paper | 70.59 | 47.35 | 2, Movie | 79.84 | 68.27 | 1, Movie | 10.89 | 8.68 |
| | 2, Paper | 40.65 | 54.44 | 1, Field | 71.44 | 82.35 | 4, Movie | 87.84 | 82.58 | 2, Movie | 22.10 | 65.56 |
| | 3, Venue | 48.51 | 72.36 | 3, Paper | 78.98 | 24.16 | 3, Movie | 91.17 | 27.24 | 4, Movie | 31.79 | 65.56 |
| | 1, Author | 50.63 | 90.74 | 3, Author | 91.52 | 91.32 | 2, Director | 91.37 | 86.57 | 4, User | 36.52 | 34.45 |
| | 4, Author | 96.01 | 18.36 | 4, Paper | 91.89 | 58.95 | 1, Movie | 91.55 | 32.33 | 1, User | 36.73 | 36.73 |
| Attention (Edge) Hu et al. (2020b) | 4, A–P | 41.54 | 18.36 | 4, P–P | 98.96 | 80.06 | 4,M–A | 98.80 | 32.63 | 4, M–U | 25.18 | 36.82 |
| | 2, A–P | 87.94 | 99.49 | 4, P–A | 91.13 | 82.19 | 3,M–A | 98.08 | 88.81 | 3, M–U | 6.23 | 76.52 |
| | 3, P–A | 90.64 | 81.77 | 3, A–P | 94.02 | 87.43 | 4,M–D | 94.69 | 84.80 | 2, M–U | 46.76 | 54.00 |
| | 3, A–P | 99.73 | 99.50 | 3, P–A | 97.71 | 94.16 | 3,A–M | 98.83 | 43.82 | 4, U–M | 6.58 | 93.70 |
| | 3, P–V | 53.30 | 72.36 | 1, P–A | 96.37 | 91.55 | 3,M–D | 100.00 | 97.59 | 2, U–M | 6.55 | 76.52 |

subsequent layers, edges *P–V* and *V–P* display the highest fidelity. Similarly, in the last layer, the edge *P–A* exhibits significantly higher fidelity, even exceeding three times the fidelity of edges *P–T* and *P–V*. In the OGB-MAG dataset, the last layer edges *P–F* and *A–I* are consistently present in top-fidelity explanations, and sequences with the *-M–D–M* suffix demonstrate high fidelity across all three types of explanations in the IMDB dataset.

In summary, the overall results underscore the potential of meta-path-level explanation to identify meaningful meta paths with sparse exclusions while aligning with existing approaches.

### 5.2.2 EXPLANATION WITH MULTIPLE PATH EXCLUSION AND COMPARISON

In addition to single path exclusion, we further investigate counterfactual examples through multiple path exclusions. Figure 4 displays the results of meta path exclusion compared to edge-level counterfactual, CF-GNNExplainer (Lucic et al., 2022), on meta structure. The experiment was conducted with distances $dist \leq 2, 3$, and 4, seeking optimal fidelity under various sparsity constraints. For comparison, the baseline was tested with a best-performed distance $dist \leq 4$.

The first notable finding is that **our methodology shows better fidelity for higher sparsity range**s. The fidelity of our meta-path-level method converges at higher sparsity than explanations without meta paths, supporting our claim that the path-based method can explain the model at a more fine-grained level, achieving similar fidelity with significantly higher sparsity. Since counterfactual explanations generally require minimal deviation from the original input, requiring high sparsity, we conclude that the proposed path-based counterfactual significantly outperforms in global graph explanations. Second, by increasing the upper bounds of distance, we could extend the sparsity range where our method is advantageous. As the distance limit increases (from Figure 7a and 4d to Figure 4c and 4f), the convergence points of fidelity and sparsity coverage both decrease. For the case $dist \leq 4$ (Figure 4c, 4f), our method shows better performance for sparsity thresholds under 80%.

In summary, the results indicate **the superior performance of meta-path-level explanations under close counterfactual conditions** compared to existing approaches. This suggests the possibility of applying more detailed analyses than existing methods.

### 5.3 APPLICATION OF META PATHS

We assessed the efficacy of our meta path counterfactual explanation by demonstrating its fidelity with higher sparsity. Extending beyond explaining the trained model, meta paths represent generally

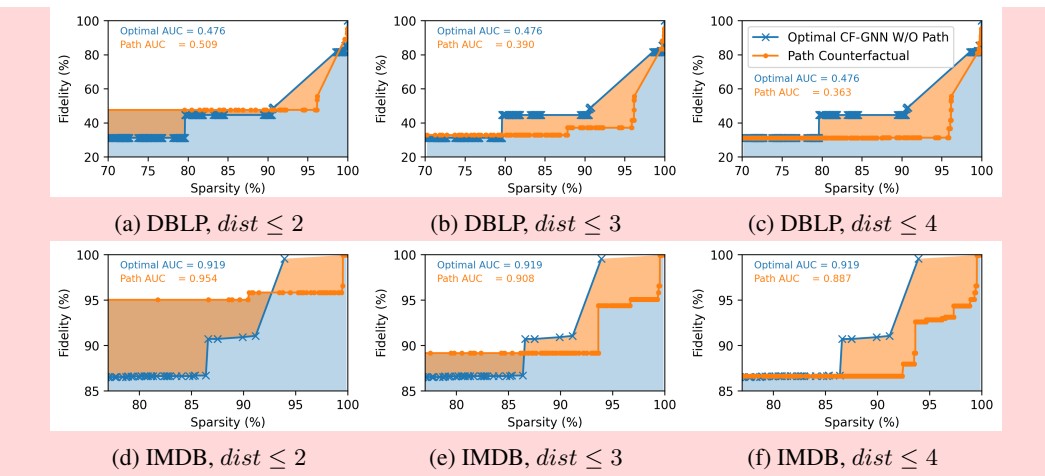

Figure 4: Plots illustrating the best fidelity for counterfactual explanations across various sparsity thresholds. Subplots (a)-(c) display the results for the DBLP dataset, while subplots (d)-(f) present the results for the IMDB dataset. For numerical comparison, each figure also reports the normalized AUC score (lower is better).

Table 2: Accuracy of meta-path-dependent HGNN model. The accuracy is evaluated when each meta path is excluded from the base model, as well as when each meta path is the sole predefined meta path. The base model is trained with meta path set derived from the original paper. The fidelity values are from the main experiment (Table 1). All values are percentages (%).

| | DBLP | | | IMDB | | |
|---|---|---|---|---|---|---|
| | Meta paths | Accuracy↑ | Fidelity↓ | Meta paths | Accuracy↑ | Fidelity↓ |
| Base Model | | 91.74 | . | | 55.87 | . |
| Excluded | -APA | 91.28 | . | -MDMDM | 55.69 | 98.54 |
| | -APAPA | 91.13 | 98.16 | -MDMAM | 56.56 | 99.94 |
| | **-APVPA** | **79.86** | **55.57** | **-MAMDM** | **55.12** | **99.88** |
| | -APTPA | 91.10 | 99.51 | -MAMAM | 56.15 | 99.97 |
| Single Path | +APA | 80.23 | . | +MDMDM | 54.95 | 98.54 |
| | +APTPV | 76.02 | 98.16 | +MDMAM | 50.55 | 99.04 |
| | **+APVPA** | **91.77** | **55.57** | **+MAMDM** | **55.06** | **99.88** |
| | +APVPT | 79.34 | 99.51 | +MAMAM | 49.80 | 99.97 |

critical information within graph models. Therefore, the objective of this subsection is to highlight the potential for identifying critical meta paths in general applications.

The lack of ground truth regarding which meta path is truly important in a dataset presents a challenge. To address this, we employ meta-path-dependent heterogeneous graph neural network (HGNN) models—known for their sensitivity to predefined meta paths—as a surrogate measure of the broader significance of meta paths. We vary the predefined meta path set used to train the accuracy of meta-path-dependent HGNN, considering the meta path set provided in a previous paper (Fu et al., 2020) as the base model. We conduct experiments on the DBLP and IMDB datasets, excluding OGB-MAG due to the unavailability of the meta path set in the original paper. For the meta-path-dependent model, we utilized the heterogeneous attention network (HAN) model.

Table 2 presents the accuracy of HAN using various predefined meta paths. The findings indicate a decrease in accuracy when excluding meta paths that exhibit high fidelity in meta-path-level counterfactual explanation. For instance, in the DBLP dataset, accuracy declines by over 10% when excluding the meta path *A-P-V-P-A*, which has the highest fidelity, from the predefined meta path set. Conversely, any other exclusion shows minimal change in accuracy. Similarly, in the IMDB dataset, the meta path *M-A-M-D-M*, which has the second-highest fidelity, results in the lowest accuracy when excluded. Moreover, when training the HAN model with only a single meta path, *A-P-V-P-A*, the top meta path in DBLP, and *M-A-M-D-M*, meta path showing the second highest fidelity in IMDB, exhibit higher accuracy than any other meta paths. Notably, the accuracy achieved solely with *M-D-M-D-M*, which has the highest fidelity, also surpasses that of others. These results suggest that the meta paths from the original paper may not represent the optimal set of meta paths

in terms of accuracy, underscoring the significance of meta-path-level explanation. It is important to note that **the brute-force search for critical meta paths incurs heavier computational costs** than our meta-path-level counterfactual explanation, as it requires retraining for each exclusion of meta path, whereas our explanation method only entails *a single forward step*.

One key point to emphasize is that our methodology is an explainable AI approach aimed at fundamentally understanding the behavior of a given trained model. Consequently, the quality of the explanations naturally depends on how well the base model itself is trained. Our underlying assumption is that **explanations extracted from a well-trained model are more meaningful and representative of the underlying heterogeneous graph data**. To validate this assumption, we track both the test accuracy and the fidelity score when excluding the *A–P–V–P–A* meta-path during the training phase of the base model on the DBLP dataset. As shown in Figure 5, the fidelity begins to drop sharply once the model accuracy exceeds a certain threshold (approximately 0.5). This trend suggests that a well-trained model (i.e., a model achieving higher accuracy) consistently identifies *A–P–V–P–A*—a meta-path previously known to be important—as critical, which is reflected by its lower fidelity when this path is removed. These observations support our assumption.

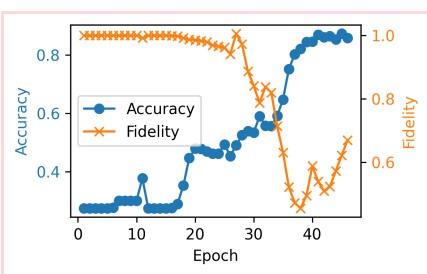

Figure 5: Illustration of the accuracy and fidelity change while training the base model on DBLP dataset excluding a meta path *A-P-V-P-A*.

## 6   CONCLUSION

As heterogeneous graph neural networks gain traction in tasks involving graph-structured data, it is imperative to understand and interpret the underlying causes of model behavior. In this study, we introduced a novel meta path counterfactual explanation approach and proposed path exclusion algorithm to selectively exclude meta paths from the computation graph of heterogeneous graph neural networks. Our findings underscore the effectiveness of meta path counterfactual explanation in elucidating model behavior with respect to granularity of explanation, characterized by high sparsity, while concurrently preserving the context of other levels of granularity. Moreover, we highlight the potential for identifying generally critical meta paths within heterogeneous graph structures through the utilization of meta-path-dependent HGNNs. Transitioning to meta-path-level explanation for explainable artificial intelligence (XAI) on heterogeneous graph neural networks promises to render explanations detailed and comprehensive, leveraging the meta path perturbation approaches we provided. We anticipate that future research efforts will focus on optimizing the explanation process for HGNNs and developing efficient methods for discovering critical meta paths.

### REPRODUCIBILITY STATEMENT

All code and experimental results are publicly available at `https://doi.org/10.5281/zenodo.11218749`. The repository includes the implementation of the path exlcusion algorithm (Algorithm 1) as a forward-function wrapper for HGT, scripts to reproduce the multi-path exclusion experiments, and the code for generating Figure 7. Detailed descriptions of datasets, ground truths, and hyperparameters are provided in the Appendix D.2. The repository will also be updated shortly with trained model checkpoints and preprocessed data to further facilitate reproducibility.

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

# A    RELATED WORK COMPARISON

Table 3: Comparison of features between existing graph explanation methods and our method. Check mark stands for compatible features in methods.

| Methods | CF. | Level | Global | Ref. |
|---|---|---|---|---|
| GNNExplainer | . | Node | . | Ying et al. (2019) |
| SubgraphX | . | Subgraph | . | Yuan et al. (2021) |
| PGMExplainer | . | Edge | △ | Vu & Thai (2020) |
| GNN-LRP | . | Walk | . | Xiong et al. (2023) |
| XGNN | . | Subgraph | ✓ | Yuan et al. (2020) |
| MEG | ✓ | Edge | . | Numeroso & Bacciu (2021) |
| Bajaj et. al. | ✓ | Edge | . | Bajaj et al. (2021) |
| CF-GNNExplainer | ✓ | Node | . | Lucic et al. (2022) |
| CLEAR | ✓ | Graph | . | Ma et al. (2022) |
| GCF-GNNExplainer | ✓ | Graph | ✓ | Huang et al. (2023) |
| Wang et. al. | ✓ | Graph | ✓ | Wang et al. (2023a) |
| CF-HGExplainer | ✓ | Node, Edge | . | Yang et al. (2023a) |
| xPAth | ✓ | Path | . | Li et al. (2023) |
| Our Method | ✓ | Meta path | ✓ | . |

# B    PATH EXCLUSION ALGORITHM

---

**Algorithm 1:** Path exclusion algorithm

---

**Input:** Heterogeneous graph $\mathcal{G} = (\mathcal{V}, \mathcal{E})$, the original HGNN model $\Phi$, initial node embedding $H$, target meta path set $P$, and target meta path tree $T_P$

**Output:** A counterfactual embedding $\tilde{\mathbf{H}}$

**begin**

  **foreach** $k \in \{1 \text{ to } d\}$ **do**

    $Conv \leftarrow \Phi_k$;

    $\mathbf{H}' \leftarrow Conv(\mathbf{H}, \mathcal{E})$;

    **if** $k=1$ **then**

      **foreach** $n \in T_P$ $s.t.$ $layer(n) = 1$ **do**

        $R_{CF} \leftarrow$ the last edges of meta paths corresponding to n;

        $\mathcal{E}_{CF} \leftarrow \{e | e \in \mathcal{E} \text{ and } \psi(e) \in R_{CF}\}$;

        $\tilde{\mathbf{H}}'_{1,n} = Conv(\mathbf{H}, \mathcal{E}_{CF})$

      **end foreach**

    **else**

      **foreach** $i \in \mathcal{V}$ **do**

        **if** $\phi(i) = A_m,\ \exists n, m\ s.t.\ m \in C(n)$ **then**

          $\tilde{\mathbf{h}}^i_{k,n} \leftarrow conv(\tilde{\mathbf{h}}^i_{k-i,m}, \mathcal{E})$;

        **else**

          $\tilde{\mathbf{h}}_{k,n}[i] \leftarrow \mathbf{h}'^i$;

        **end if**

      **end foreach**

    **end if**

    $\mathbf{H} \leftarrow \mathbf{H}'$;

  **end foreach**

  **return** $\tilde{\mathbf{H}}_{d,root}$;

**end**

---

## C    THE USE OF LARGE LANGUAGE MODELS

We used ChatGPT as a writing assistant to polish the language of this paper, such as improving clarity, grammar, and readability. The LLM was not involved in the research design, ideation, experiments, analysis, or in drawing any scientific conclusions.

## D    DETAILED EXPERIMENTAL SETTINGS

### D.1    METRICS

We define several metrics for evaluating global counterfactual explanations on meta structures. Since ground-truth explanations are not available for most datasets, metrics based on distance from the ground truth (e.g., accuracy or F1 score) cannot be used. Instead, we assess the importance of counterfactual examples by measuring the proportion of data points whose predictions differ from the original prediction after applying the counterfactual. In addition, we assess explanation sparsity by calculating the proportion of excluded data instances within the generated explanations.

**Fidelity.** For global counterfactual explanations, we propose using fidelity as a substitute for the ground truth accuracy of the explanations. We adapt the definitions in GraphFramEx (Amara et al., 2022) to focus on global counterfactual explanations for the meta structure of heterogeneous graphs. According to the Global CF-GNNExplainer (Huang et al., 2023), the main optimization goal of the global recourse of GNN is to regulate as many negative samples as possible using the counterfactual example (in this case, a set of recourse graphs), known as coverage, while maintaining a certain distance between the recourses and data points. We extend this definition to include both negative and positive samples, aiming to consider both the correctness and errors of the model. Instead of a set of graphs, we form a single perturbed meta structure as a counterfactual example. This approach aligns with the concept of fidelity.

$$fid = \frac{1}{N} \sum_{i=1}^{N} \mathbb{1}(\hat{y}_i^{G_i/P} = \hat{y}_i) \tag{5}$$

**Distance and Sparsity.** Even if an explanation flips predictions for a large portion of the dataset, it cannot be considered a good counterfactual example unless the perturbation is small. High-quality counterfactuals should be both **effective** (changing the prediction) and **sparse** (minimally different from the original input). To capture this trade-off, we use two metrics: *distance* and *sparsity*. Distance measures how far the counterfactual example is from the original input. In our framework, this is quantified as the number of perturbed meta paths ($P$). Sparsity captures how much of the input graph component has been unaltered. Following the standard definition in Yuan et al. (2022), sparsity is defined as the average fraction of unperturbed elements:

$$dist = \mathcal{N}(P), \quad Spar = 1 - \frac{1}{N} \sum_{i=1}^{N} (\frac{m_i}{M_i}) \tag{6}$$

where $m_i$ is the number of perturbed features for instance $i$, and $M_i$ is the total number of features. In this work, we adopt meta path instances as the input features of interest, treating them as the atomic semantic units of HGNN reasoning.

### D.2    BASELINES

As there are no existing explainability methods directly applicable to global and meta-path-level explanations in HGNNs, we adapt several influential local explainability methods to the meta-structure setting. These adaptations serve as baselines for evaluating our approach. **CF-GNNExplainer** (Lucic et al., 2022) generates edge-level counterfactual explanations by perturbing the adjacency matrix of local graph instances. To adapt this to the meta structure, we implement edge type removal at specific layers, identifying the layer–edge-type pairs whose exclusion yields the highest fidelity. **PGM-Explainer** (Vu & Thai, 2020) provides node-level explanations via feature perturbation. We adapt it by replacing the features of a specific node type in the meta structure with the average feature vector of that type, approximating node-level perturbations in the meta context. **Attention**-based explanation, as suggested in the original HGT paper (Hu et al., 2020b), interprets attention

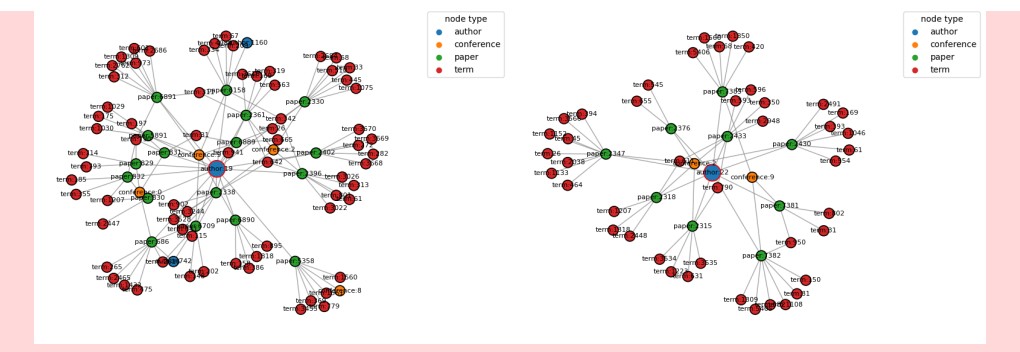

Figure 6: Ego-network examples in which the predicted label of the center author node flips when the *A–P–V–P–A* meta-path is removed by our algorithm. The figures illustrate that these authors typically publish in only a small number of venues and exhibit numerous 2-hop connections with terms, highlighting the strong influence of venue information on classification.

weights as indicators of importance. We compute the largest attention value for each node type at each layer and use this as an implicit importance score.

These adapted baselines allow us to compare different granularity levels (node, edge, and meta path) under a common meta-structural framework, highlighting the fine-grained interpretability of our proposed method.

### D.3 DATASETS

We selected three different real-world heterogeneous graph datasets focusing on node classification tasks: DBLP (Fu et al., 2020), OGB-MAG (Hu et al., 2020a), and IMDB (Fu et al., 2020).

**DBLP** is an academic graph consisting of four node types: author, paper, venue, and term. The main task is to predict the authors' research areas among four divisions: database, data mining, artificial intelligence, and information retrieval.

**OGB-MAG** is another academic graph from Microsoft, including author, paper, field of study, and institute. Unlike DBLP, the task is to classify the paper nodes' venue.

**IMDB** is a heterogeneous graph based on movie information, containing three node types: movie, director, and actors. The task is to predict the movies' genre from action, comedy, or drama.

**MovieLens** is a relatively simple heterogeneous graph dataset constructed from movie rating information. It contains two node types—*users* and *movies*—and is formulated as an edge classification task, where each edge corresponds to a user–movie interaction and the user's rating serves as the label for that edge.

## E FURTHER ANALYSES

### E.1 QUALITATIVE STUDY

In this section, we present a case study on the DBLP ego network to illustrate how removing specific meta paths, particularly those identified as important by our method, affects node classification in a real-world academic graph. To examine the impact of the most critical meta-path, *A–P–V–P–A*, we sampled authors whose predicted labels flip when this meta-path is excluded during explanation. Our analysis (Figure 6) reveals that these authors typically publish in only a small number of venues (conferences) and maintain a large number of 2-hop connections with terms. This suggests that venues contain rich information about an author's publications and can strongly indicate the author's research domain. In contrast, the *A–P–T–P–A* meta-path appears relatively noisy: only a small set of terms is consistently shared across an author's papers, whereas venues are often shared

across nearly all of an author's publications. These observations explain why *A–P–V–P–A* is particularly influential in the DBLP dataset. It effectively captures the semantic relationship of "authors publishing in the same venue," which serves as a strong signal for identifying an author's research direction.

### E.2 GENERALIZABILITY ON OTHER BASE MODELS

To further validate the model-agnostic nature of our method, we have conducted additional experiments using the GAT aggregation on HGT model as another meta-path-independant HGNN base model. The results exhibit a similar trend to those observed with normal HGT, further supporting the generalizability of our approach. For the DBLP dataset, the *Author-Paper-Venue-Paper-Author* meta path achieves a fidelity score of 44.60%, which is significantly lower than other meta paths, such as *Author-Paper-Venue-Paper-Term* and *Author-Paper-Venue-Paper-Venue*, both of which have fidelity values around 0.7. Furthermore, for meta paths deemed insignificant, such as certain random paths, there are no observed label flips when excluding them, underscoring their lack of influence on the model's behavior. For the IMDB dataset, the meta-path *M-A-M-D-M*, which ranks second in fidelity for the HGT model, achieves the lowest fidelity in this model with a fidelity score of 90.45%. This consistency aligns well with the trends reported in Table 3 of our manuscript.

These results demonstrate the robustness of our method across different HGNN architectures. We will include these additional experimental results in the revised manuscript to substantiate our claim of model agnosticism.

### E.3 COMPARISON WITH ATTENTION

As a qualitative assessment, we present visualizations to illustrate the disparity between our method and explanation by attention approaches.

Figure 7a compares the fidelity between our method and the attention values of the HGT model. Previous literature (Hu et al., 2020b) suggests that latent attention values from the HGT model may reveal importance of certain meta paths. To investigate this, we extracted attention values from the same HGT model used for our explanation and compared them with the fidelity of the meta-path-level explanation.

In the experiment on the DBLP dataset, our explanation identified the meta path *A-P-V-P-A* as the most fundamental, while attention values highlighted the *PT* edge in the first layer of the model. As shown in the preceding subsection, meta paths ending with the *PT* edge exhibit low fidelity in a counterfactual setting This discrepancy questions the reliability of using attention values as explanations, particularly in heterogeneous graph neural networks. This observation aligns with other reports indicating discrepancies between explanation methods and attention values (Wiegreffe & Pinter, 2019).

### E.4 SCALABILITY OF PATH EXCLUSION ALGORITHM

Theoretically, the time and space complexity of our algorithm is sublinear relative to the complexity of the original model's forward function. For each layer, the target meta path tree $T_P$ contains fewer nodes than $dist$, where $dist$ is the number of perturbed meta paths. Therefore, the maximum number of augmented calculations in path exlcusion algorithm is bounded by $dist$. As a result, the worst-case time and space complexity is $O(dist)X$, where X represents the original forward complexity. On average, the number of nodes in $T_P$ is $log(dist)n$, where n is number of layers. Consequently, the average complexity becomes $O(log(dist))X$. This indicates that our algorithm remains practical, enabling efficient monitoring of specific path exclusions.

Additionally, we conducted empirical evaluations to measure forward time consumption under varying numbers of excluded paths. The results (Figure 7b) demonstrate sublinear growth across all datasets, with data points consistently within a $\pm 0.02$ margin from the average, validating the above claim.

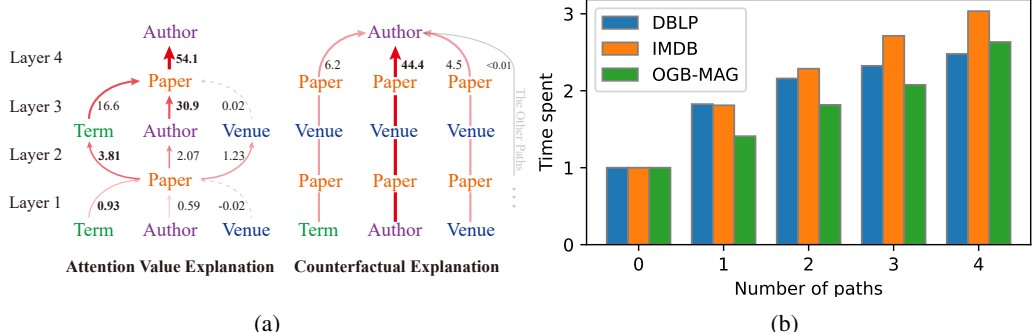

(a)                                    (b)

Figure 7: (a) Visualization of attention values of edge type of HGT and meta-path-level explanation by our method. (b) The forward time consumption of path exclusion algorithm evaluated under varying numbers of target paths, scaled relative to the original forward time.

### E.5 FURTHER DISCUSSION

**General Applications.** A common use of model explanations is to refine and enhance models based on the insights gained. This principle also applies to meta-path-level global explanations for HGNNs. Beyond the accuracy improvements observed in meta-path-dependent models (in Section 5.3), another potential benefit is data augmentation based on critical meta paths. When a specific meta path is deemed crucial for a model, it serves as a signal to collect more data that follows such meta structure. For example, in the DBLP dataset, an HGNN operator might discover that certain *author–venue–author* relationships significantly impact the model's performance by meta path *A–P–V–P–A*. Instead of merely expanding venue-related information, operators can focus on gathering more data about authors publishing in those venues, leading to better model performance and interpretability.

**Domain-specific Problems.** We conducted a case study in cybersecurity domain using the provenance graph dataset named Streamspot[2], which represents system logs as heterogeneous graphs. Applying our method to attack graph detection, we identified a low-fidelity (0.15–0.6), high-sparsity meta path: PROCESS-[mmap2] $\rightarrow$ MAP_ANONYMOUS $\leftarrow$ [mmap2]—THREAD—[send]$\rightarrow$socket. This meta path can be interpreted as two distinct processes and threads mapping MAP_ANONYMOUS and connecting to other sockets, which correlates with a drive-by-download attack (Manzoor et al., 2016). This demonstrates the utility of meta-path-level explanations in two key ways: 1. Structural relevance: The meta structure is critical, as it captures behavior patterns that generalize across instances.s 2. Path information: Series of events connections are critical, as neither node nor edge types alone could capture this complex pattern. These results illustrate how domain experts can leverage meta-path-level explanations to identify attack patterns and inform targeted interventions.

**Compatibility with Optimization Techniques.** This work introduces a novel approach for counterfactual explanation in heterogeneous graphs by perturbing meta paths—without modifying the original graph. Unlike prior studies that focus on identifying optimal explanation sets, our objective is not to maximize fidelity but to establish meta paths as a fine-grained and semantically meaningful unit of explanation. We emphasize atomic exclusions, demonstrating that our method can isolate critical substructures that are as influential as coarser units (e.g., nodes or edges), thereby enabling higher-resolution interpretability. To assess further effectiveness, we tested the method's compatibility with existing optimization strategies. Using a simple greedy best-first search with top-$k$ memorization, we found that the highest-fidelity meta path could consistently be identified within 12 iterations across datasets. Notably, in DBLP, the optimal path $A—P—V—P—A$ was selected in the first iteration, with fidelity effectively guiding the search at each step. These results suggest that our path exclusion algorithm can be seamlessly integrated with more sophisticated optimization techniques in future work, potentially enhancing the scalability and effectiveness of meta-path-level counterfactual explanation.

---

[2]https://github.com/sbustreamspot/sbustreamspot-data

