# OpenReview forum: "Global Meta-path-level Counterfactual Explanation for Heterogeneous Graph Neural Networks by Path Exclusion"
_ICLR.cc/2026/Conference — Submitted to ICLR 2026_

### Official Review · Reviewer_hBoM · 2025-10-28

**Soundness:** 3
**Presentation:** 2
**Contribution:** 2
**Rating:** 4
**Confidence:** 2

**Summary:**

The authors introduce a method to generate global counterfactual explanations (CFE) of heterogeneous graph neural networks. They argue that perturbing nodes, edges, or other structural features cannot satisfactorily give high quality CFE because each perturbation of this nature can fail to identify the phenomenon that actually dictates the model behavior. They propose a framework that is a post-hoc explainer that perturbs meta paths instead to generate CFE for HGNN. They use their spare path algorithm to exclude the computation of the leaf part of the excluding path in the HGNN on the forward pass. After perturbed path embeddings are computed the framework optimizes to find the optimal meta path (which acts as a CFE). They conduct experiments with relevant baselines on fidelity, distance, and sparsity. Their initial experiments suggest improvements when using their framework over existing

**Strengths:**

S1. The paper is well written, simple, logical and provides a novel angle to solving a niche problem; global CFE on HGNNs.


S2. Methodology is reasonable and initial implementation decisions make sense.


S3. The results in some experiments suggest that the method is more effective than existing global graph CFE methods particularly when applied to meta paths explanations.

**Weaknesses:**

W1. The work provides a reasonable solution to an overlooked problem. Although the experiments suggest their methodology can exhibit superior performance the results are by no means definitive. They need more experiments to supplement their claims.
W2. The authors leave a lot of technical details out of the main body of the paper that are necessary. For instance, even though the optimization procedure is not novel to the work the authors fully leave out how they optimize the best perturbed meta path. They state this is put in the appendix but there is a serious lack of technical details in the paper that are quite necessary.

**Questions:**

1) I believe as per W1 more experiments are needed to justify the superiority of this method. If you can include some additional experimental datasets (this is necessary to bolster claims) and possibly some additional baselines (if needed).
2) The paper should clearly indicate the technical details of their methodology. These details should not be delegated to the appendix. It should be clear how the framework works in all of its technical details.

---

> ### Author Response · Authors · 2025-11-21
>
> We sincerely thank the reviewer for the constructive and detailed feedback. Below we address each point and describe the corresponding revisions that will be incorporated into the updated manuscript, which we will upload early next week.
>
> 1. Need more experiments to supplement their claims.
>
>     We agree that the fine-grained nature of our explanation method should be demonstrated more clearly. To reinforce this claim, we have conducted several new experiments:
>
>     - **Additional Dataset (MovieLens):**
>
>         We introduce a new link-prediction dataset, **MovieLens**, which has a simpler relational structure. In this dataset, the meta-path **UMUMU** achieves high fidelity with extremely low sparsity (0.12), highlighting the fine-grained behavior of our meta-path–based explanation compared to baselines. A summary of key results is shown below; full results will appear in the final manuscript. These results show that our method identifies explanations with substantially lower sparsity than existing edge-centric baselines.
>
>         | Explanation | Fidelity | Sparsity |
>         | --- | --- | --- |
>         | UMUMU | 64.81 | 0.12 |
>         | MUMUM | 34.20 | 31.22 |
>         | 1, M-U | 94.04 | 6.30 |
>         | 4, U-M | 93.42 | 6.30 |
>         | 2, Movie | 77.90 | 34.44 |
>         | 4, Movie | 68.21 | 34.44 |
>     - **Edge-level explanation analysis:**
>
>         In addition, we will include a new table evaluating the correspondence between **edge-level perturbations** and **affected meta-paths**, analyzing sparsity correlations between our approach and existing baseline methods. This further supports our claim regarding fine-grained interpretability.
>
>
>     These expanded experiments will appear in the revised manuscript.
>
> 2. Technical details
>
>     Our primary focus is the proposed solution for **meta-path omission**, with meta-path optimization left as future work. Nevertheless, we agree that more implementation details will improve clarity.
>
>     - **Meta-path selection strategy:**
>
>         For the datasets used in the paper, we select meta-paths using **brute-force search** based on fidelity. For simpler datasets such as IMDB and DBLP, we additionally evaluate a **greedy best-first search** strategy. These methodological details will be added to the main text.
>
>     - **Model hyperparameters and training details:**
>
>         We will expand the experimental section to report all hyperparameters and training configurations. For example, on the DBLP dataset, we use:
>
>         - **4 HGT layers**,
>         - **256 hidden channels**,
>         - **learning rate of 0.0005**,
>         - constant feature vectors for *venue* nodes, and
>         - learned HGT embeddings for *author* nodes at the first layer.
>
>     These additions will improve transparency and reproducibility.

---

### Official Review · Reviewer_HWkH · 2025-10-31

**Soundness:** 3
**Presentation:** 3
**Contribution:** 3
**Rating:** 6
**Confidence:** 4

**Summary:**

This paper presents a novel framework for global meta-path-level counterfactual explanation in heterogeneous graph neural networks. The authors propose a spare path algorithm that directly excludes specific meta-paths during forward propagation to evaluate their global impact. The method is implemented as a wrapper over HGT-like architectures and evaluated on three benchmark datasets (DBLP, OGB-MAG, IMDB). Results demonstrate that the approach effectively identifies semantically meaningful meta-paths and achieves competitive fidelity–sparsity trade-offs.

**Strengths:**

1.	Conceptual novelty: Introducing meta-path-level counterfactual analysis fills a gap between edge-level and subgraph-level explainers, offering a new abstraction for heterogeneous graphs.
2.	Practical implementation: The proposed spare path mechanism is elegant, simple to integrate into existing GNNs, and supported by clear pseudo-code.
3.	Empirical support: Comprehensive experiments on multiple datasets demonstrate the method’s interpretability and scalability within realistic heterogeneous settings.
4.	The framework could benefit practical applications (e.g., academic network analysis, recommender reasoning, and knowledge graph inspection) where meta-path semantics are essential.

**Weaknesses:**

1.	Results (fidelity, sparsity, accuracy) are reported as single values without variance or confidence intervals. Including results over multiple random seeds would make the conclusions statistically stronger.
2.	The current set of baselines omits some recent counterfactual explanation approaches such as CFExplainer [1] and GCFExplainer [2].

[1] Graph neural networks for vulnerability detection: A counterfactual explanation. 2024

[2] Global counterfactual explainer for graph neural networks. 2023

3.	The adaptation of local explainers (e.g., CF-GNNExplainer, PGExplainer) to the global setting lacks detailed parameter descriptions, which slightly affects fairness assessment.
4.	The interpretation of results could be more analytical. For example, why certain meta-paths cause stronger performance drops or how redundancy between paths affects importance ranking.

**Questions:**

Please address the weaknesses.

---

> ### Author Response · Authors · 2025-11-21
>
> We sincerely thank the reviewer for the constructive and detailed feedback. Below we address each point and describe the corresponding revisions that will be incorporated into the updated manuscript, which we will upload early next week.
>
> 1. Results (fidelity, sparsity, accuracy) are reported as single values
>
>     We appreciate the reviewer pointing out the lack of variance reporting. We have **re-run all main experiments (including Table 1 and Table 2) across multiple random seeds (e.g., 5 or 10 seeds)**. In the revision, we will:
>
>     - Report **mean ± standard deviation** for Fidelity, Sparsity, and Accuracy in all relevant tables.
>     - Add a dedicated subsection in the Appendix detailing all experimental settings, including random seeds, train/validation/test splits, hardware specifications, and hyperparameters.
>
>     This ensures statistical rigor and full reproducibility.
>
> 2. The current set of baselines omits some recent counterfactual explanation approaches such as CFExplainer [1] and GCFExplainer [2].
>
>     We appreciate the reviewer pointing out these recent counterfactual explanation methods. However, we believe they are not directly applicable to our setting:
>
>     - **CFExplainer [1]** is a domain-specific adaptation of edge perturbation designed for code-graph analysis.
>     - **GCFExplainer [2]** relies on three types of perturbations: addition, removal, and label modification at the node/edge level.
>
>     As noted in Section 3.2 of our manuscript, **meta-path–level modification is inherently limited to deletion**, because addition or substitution of meta-paths is not well-defined in heterogeneous graph meta structures. Thus, these recent methods fall under the broader umbrella of edge-perturbation approaches and are effectively covered by our chosen baselines (Lucic et al. and Vu & Thai). We will clarify this point more explicitly in the revised manuscript to avoid misunderstanding.
>
> 3. The adaptation of local explainers (e.g., CF-GNNExplainer, PGExplainer) to the global setting lacks detailed parameter descriptions, which slightly affects fairness assessment.
>
>     In our experiments, we adopt an intentionally **stricter evaluation setting**: for each baseline, we replace its local optimization component with a **brute-force search for the global optimum**. This ensures that: the comparison is *at least as favorable* to the baselines as their default configuration, and our evaluation is compared with the **theoretically optimal performance** each baseline can achieve, rather than the performance of their original heuristic or local-search procedures.
>
> 4. The interpretation of results could be more analytical.
>
>     We acknowledge that our interpretative analysis can be expanded. As part of the revision, we are conducting additional qualitative evaluation. Section 5.2 will be revised to provide deeper interpretation of the discovered meta-paths. For example, we will elaborate on why **A–P–V–P–A** is particularly influential in DBLP: it captures the semantic relationship of *authors publishing in the same venue*, a strong indicator of academic field classification.
>
>     To fully address the reviewer’s suggestion, we will also add a **Qualitative Case Study** to the Appendix, including a visual depiction of explanations on a real subgraph from the anonymized dataset.

---

### Official Review · Reviewer_SWZT · 2025-11-01

**Soundness:** 2
**Presentation:** 1
**Contribution:** 3
**Rating:** 4
**Confidence:** 3

**Summary:**

Counterfactual explainability in heterogeneous graphs is a challenging problem. The paper introduces the Spare Path algorithm, which removes meta-paths while preserving the remainder of the heterogeneous graph. Evaluated on three datasets, the method demonstrates strong effectiveness.

**Strengths:**

- Addresses a hard, real‑world problem with clear practical relevance.
- Code is available to support reproducibility, though some components appear incomplete.
- Results are promising but limited in scope and breadth.
- Method is simple yet effective, offering a straightforward baseline with strong initial performance.

**Weaknesses:**

- The paper is poorly written; several sections are difficult to follow and transitions between sentences are weak.
- In Table 2, meta-path exclusion or usage has little impact on IMDB but shows some effect on DBLP.
- The experimental evaluation is underpowered, with too few datasets.
- There is no ablation study and no concrete procedure for meta-path selection.

**Questions:**

- Can you validate the Table 2 findings on additional datasets to test whether the meta-path effect generalizes?
- Please expand the experimental suite with more heterogeneous graph datasets to better demonstrate the model’s effectiveness.
- How many independent runs did you perform? Provide full experimental settings (seeds, splits, hardware, hyperparameters) and report mean -+ standard deviation across runs. If results are from a single run, please rerun multiple times and add variance metrics and statistical significance.
- Is there a principled way to select meta-paths a priori; without inspecting outcomes from the Spare Path algorithm (e.g., via heuristics, validation criteria)?

---

> ### Author Response · Authors · 2025-11-21
>
> We sincerely thank the reviewer for the constructive and detailed feedback. Below we address each point and describe the corresponding revisions that will be incorporated into the updated manuscript, which we will upload early next week.
>
> 1. On Writing Quality and Baseline Adaptation Details
>
>     We acknowledge the issues with clarity and organization. In the revised manuscript, we will thoroughly revise the writing to improve readability, coherence, and overall presentation.
>
> 2. Table 2, meta-path exclusion or usage has little impact on IMDB but shows some effect on DBLP
>
>     Regarding the **IMDB results in Table 2**, we agree that excluding a single meta-path produces a smaller effect than in DBLP. Our interpretation is that two meta-paths, **MDMDM** and **MAMDM,** are both highly influential for the IMDB model. Because these two paths individually exhibit strong standalone predictive performance, removing only one of them does not lead to a substantial accuracy drop. This observation is consistent with our fidelity scores, where these two meta-paths rank highest.
>
>     We also emphasize that **explanations are model-dependent**, i.e., the explanation reflects the behavior of the trained model rather than an intrinsic property of the dataset. In Table 2, our assumption is that the underlying HGNN is well-trained; however, this may not always hold. To further investigate this, we have additionally analyzed performance during the training process by tracking both model accuracy and the fidelity of the APVPA meta-path throughout training. This reveals how the model identifies critical meta-paths over time. These results will be included in the revised manuscript.
>
> 3. Limited number of datasets
>
>     To address concerns about dataset diversity, we have added a new link prediction dataset, **MovieLens**, which provides a simpler relational structure. In this dataset, the meta-path **UMUMU** yields high fidelity with extremely low sparsity (0.12), highlighting that our meta-path–based explanations are more fine-grained than baseline approaches. These results further validate the applicability and fine-grained behavior of our approach across multiple datasets.
>
>     A summary of key results is shown below; the full table and discussion will appear in the revised manuscript:
>
>     | Explanation | Fidelity | Sparsity |
>     | --- | --- | --- |
>     | UMUMU | 64.81 | 0.12 |
>     | MUMUM | 34.20 | 31.22 |
>     | 1, M-U | 94.04 | 6.30 |
>     | 4, U-M | 93.42 | 6.30 |
>     | 2, Movie | 77.90 | 34.44 |
>     | 4, Movie | 68.21 | 34.44 |
> 4. Meta-path selection
>
>     Our work focuses on *evaluating* the effect of omitting specific meta-paths, and we leave the design of optimized selection algorithms to future research. For the datasets discussed in the paper, we find optimal fidelity using brute-force search, and we additionally evaluated a simple greedy best-first search strategy for IMDB and DBLP. We will expand the manuscript to clearly describe these selection procedures.
>
> 5. Ablation study
>
>     The proposed spare-path (path-exclusion) algorithm is a single-module procedure and cannot be decomposed into multiple independently meaningful components. For this reason, a traditional ablation study is not applicable. We will clarify this point in the revision.
>
> 6. Rerun multiple times and add variance metrics and statistical significance.
>
>     We appreciate the reviewer pointing out the lack of variance reporting. We have **re-run all main experiments (including Table 1 and Table 2) across multiple random seeds (e.g., 5 or 10 seeds)**. In the revision, we will:
>
>     - Report **mean ± standard deviation** for Fidelity, Sparsity, and Accuracy in all relevant tables.
>     - Add a dedicated subsection in the Appendix detailing all experimental settings, including random seeds, train/validation/test splits, hardware specifications, and hyperparameters.
>
>     This ensures statistical rigor and full reproducibility.
>
> 7. principled way to select meta-paths
> For meta-path–dependent HGNNs, there is currently **no widely adopted principled or algorithmic method** for selecting meta-paths prior to training. Although a few works study meta-path identification from data, these approaches are *not* explanation methods and do not aim to interpret the behavior of a trained model. Our method instead analyzes how an already trained model relies on specific meta-paths, which serves a fundamentally different purpose. We will clarify this distinction in the revised manuscript.

---

### Official Review · Reviewer_yj6y · 2025-11-02

**Soundness:** 2
**Presentation:** 3
**Contribution:** 2
**Rating:** 4
**Confidence:** 4

**Summary:**

This paper proposes a framework for global meta-path-level counterfactual explanations in heterogeneous graph neural networks (HGNNs). The main contribution is an algorithm that excludes specific meta paths during the forward pass to estimate their influence on model predictions. The goal is to achieve finer-grained and semantically meaningful explanations compared to traditional node- or edge-level perturbations. Experiments on DBLP, OGB-MAG, and IMDB show that removing a small number of meta paths can significantly degrade accuracy, suggesting that these paths are crucial for model reasoning.

The topic is timely, and the conceptual novelty is evident. However, the paper in its current form suffers from critical flaws in its evaluation methodology, including inconsistent metric definitions and unclear experimental presentation, which undermine the validity of its conclusions.

This paper should be rejected because (1) the definitions and application of core evaluation metrics are flawed, reversing the established meanings of fidelity and sparsity from cited work without justification, (2) the reported fidelity and sparsity values contradict their theoretical range, which undermines the credibility of the quantitative results, (3) the presentation of experiments is difficult to understand, particularly in Table 1, which lacks the necessary clarity for interpretation, and (4) the analysis of results is superficial, failing to provide sufficient interpretive depth to convincingly support the paper's conclusions.

# Overall Evaluation

This is a creative and promising submission introducing a new perspective on HGNN explainability. However, the work is undermined by inconsistent and non-standard metric definitions, unclear reporting in tables, and a discussion that lacks interpretive depth. With clear terminology, explicit and correct metric definitions, improved table formatting, and a deeper analytical discussion, the paper could become a valuable contribution in future iterations. In its current state, the foundational issues with the evaluation prevent a confident assessment of the method's effectiveness.

**Strengths:**

The idea of counterfactual reasoning at the meta-path level is original and well-motivated. The implementation of the proposed path exclusion algorithm appears technically sound and is clearly explained. The approach has the potential to open a promising new direction for HGNN interpretability once the significant issues with evaluation and presentation are addressed.

**Weaknesses:**

A critical flaw lies in the definition, interpretation, and reporting of the evaluation metrics. The paper cites fidelity and sparsity from prior work (CF-GNNExplainer and Yuan et al., 2022), but inverts their standard interpretations: fidelity is treated as higher-is-better and sparsity as lower-is-better, which is the opposite of the cited definitions. This fundamental inversion is not justified or even acknowledged in the main text, causing significant confusion and making it difficult to compare results against the established literature.

Compounding this issue, both reported fidelity and sparsity values frequently exceed 1 (e.g., values like 44.43 for fidelity and 81.64 for sparsity in Table 1), contradicting the theoretical [0, 1] range for such metrics. These appear to be percentages, but this is never explicitly stated. Even if they are percentages, this does not resolve the core problem of the inverted definitions and makes the quantitative results difficult to trust. A rigorous and consistent evaluation framework is essential for the paper's claims to be verifiable.

Table 1 is particularly unclear and requires substantial revision. To make it interpretable:

Add arrows in the column headers (Fidelity~↑, Sparsity~↓) to explicitly indicate the optimization direction you are assuming.
Use boldface to highlight the best-performing result within each dataset and metric column.
Clarify in the caption that the Accuracy row at the top of each dataset refers to the baseline performance of the HGNN before any meta-path exclusion and is not tied to a specific path explanation.
Include a short note or footnote specifying that Fidelity and Sparsity values are reported as percentages and clarify their definitions.

The discussion of results is largely descriptive, with limited interpretation of why certain meta paths are more influential than others. Captions merely restate what figures depict instead of summarizing insights. As a result, the empirical analysis does not convincingly support the conclusions drawn in the text.

**Questions:**

# Clarifications on Current Presentation
1. Terminology: Please clarify the precise meaning of “spare path” and how it differs conceptually and computationally from a more straightforward term like “excluded path”.
2. Metrics: Why do fidelity and sparsity values exceed 1 in Table 1? Please confirm if they are expressed as percentages and, more importantly, justify the inversion of their standard definitions from the literature. Furthermore, the evaluation would be strengthened by including the performance of the comparison models on the actual paths. Please provide these results or explain their omission.
3. Table 1 Clarity: Could you expand the description of Table 1 to explicitly link the “Accuracy” row to the baseline model and to specify which metrics are improved by higher or lower values?


# Suggestions for Additional Experiments

1. Sparsity Correlation: To better ground your sparsity metric, could you provide analyses connecting it to simpler graph perturbation statistics? For instance, how does the meta-path sparsity of an explanation correlate with the total number of unique edges removed or the average change in node degree across the graph? This would help validate the claim of achieving "fine-grained" explanations.
2. Fidelity-Sparsity Trade-off: The trade-off between explanation quality (fidelity) and cost (sparsity) is central to your work. Could you provide AUROC curves or correlation plots for fidelity versus sparsity? This would offer a more standardized and quantitative way to compare your method against baselines than the current plots in Figure 4.
3. Qualitative Case Study: To address the lack of interpretive depth, could you include a qualitative case study? For example, for the most important meta-path found (e.g., A-P-V-P-A), show a concrete subgraph where removing its instances flips a node's prediction and provide a domain-specific reason why this occurs.
4. Model Robustness: The experiments are performed on HGT. How robust are the identified important meta-paths to the choice of the underlying HGNN model? Showing that similar meta-paths are identified as important using a different architecture (e.g., HAN) would significantly strengthen the claim that your method uncovers fundamental data patterns rather than model-specific artifacts.

# Minor comments

In Fig 3, there appears to be a typo in the label of the third figure. Furthermore, there are two em dashes in the first paragraph, probably mistyped between two dashes.

---

> ### Author Response · Authors · 2025-11-21
>
> We sincerely thank the reviewer for the constructive and detailed feedback. Below we address each point and describe the corresponding revisions that will be incorporated into the updated manuscript, which we will upload early next week.
>
> 1. Terminology, Metrics, and Table Clarity
>
>     We appreciate the reviewer’s comments regarding metric interpretation. We have revised the definition based on the standard definitions used in the cited literature. All **Fidelity** and **Sparsity** values reported in Table 1 are **percentages (%)**, and we acknowledge that this was omitted in the original submission. In the revised manuscript, we will explicitly state in the caption that all values are percentages. We will also add directional indicators, **Fidelity (↓)** and **Sparsity (↑)**, to clarify optimization direction.
>
>     We will additionally apply **boldface** to highlight the best-performing results for each metric within each dataset.
>
>     The caption will also clearly state that the “Accuracy” row (e.g., “DBLP (Accuracy 86.74%)”) refers to the baseline accuracy of the original, unperturbed HGNN model, not the outcome of any individual explanation method.
>
> 2. Meaning of spare path
>
>     We used the term *spare path* to indicate that “the algorithm spares path embeddings over counterfactual embeddings.” However, we agree that the terminology may cause confusion. In the revision, we will rename the method to the **path-exclusion algorithm for HGNNs**, which more directly reflects its purpose and behavior.
>
> 3. Sparsity Correlation
>
>     In the DBLP example provided in the manuscript, perturbing a P–V edge at the third layer (corresponding to the A–P–V–P–A meta-path) ultimately influences three specific meta-paths in a non–self-loop setting: A–P–V–P–A, A–P–V–P–V, and A–P–V–P–T.
>
>     This result is favorable for edge-based explanation approaches because such fine-grained layer-wise propagation effects are typically not captured by them. Under a more realistic perturbation scenario involving a P–V edge, the perturbation corresponds to six meta-paths.
>
>     As the reviewer suggested, this illustrates the fine-grained nature of our method. To make this clearer, we will include a new table in the revised manuscript that reports:
>
>     - the mapping between edge-level perturbations and affected meta-paths,
>     - sparsity values for each, and
>     - correlations between sparsity and other graph statistics (e.g., edge instance counts, node degrees).
> 4. Fidelity-Sparsity Trade-off
>
>     Figure 4 in the current version already presents the **minimum-fidelity-over-sparsity curve**, equivalent to the **Area Under the Fidelity-Sparsity Curve**  when the modified metric is applied. The raw fidelity-sparsity relationship, when plotted directly, results in more cluttered visualizations. However, we agree with the reviewer that including this information can provide additional insight. We have therefore overlayed a **transparent raw fidelity-sparsity plot** on top of the existing figure in the revised manuscript.
>
> 5. Qualitative Case Study
>
>     Because our experiments use anonymized datasets, we are currently running an additional qualitative analysis. In the revision, we will expand Section 5.2 to provide a deeper interpretation of the identified meta-paths. For example, we will explain why **A–P–V–P–A** is particularly influential in DBLP: it captures the semantic relationship of “authors publishing in the same venue,” which is strongly correlated with research-field classification.
>
>     To fully address the reviewer’s suggestion, we will also include a **Qualitative Case Study** in the Appendix, presenting a visual explanation on a real subgraph from the anonymized dataset.
>
> 6. Model Robustness
>
>     To further validate the model-agnostic nature of our approach, we conducted additional experiments using the custom **GAT** aggregation HGT model (sorry for confusion). The results exhibit trends consistent with those observed for HGT:
>
>     - **DBLP dataset:**
>
>         The **APVPA** meta-path achieves a fidelity of **65.40**, substantially higher than other meta-paths such as **APVPT** and **APVPV**, both of which have fidelity values around **0.3**. Meta-paths deemed insignificant (including random ones) produce no label flips when excluded, confirming their minimal influence.
>
>     - **IMDB dataset:**
>
>         The meta-path **MAMDM**, which ranks second in fidelity for the HGT model, achieves the highest fidelity under HAN (**19.55**), aligning with the patterns reported in Table 3 of our manuscript.
>
>
>     These results support the robustness and model-agnostic generalizability of our method. We will incorporate details of these findings into the revised manuscript.

---

> > ### Comment · Reviewer_yj6y · 2025-11-24
> > **Thanks**
> >
> > Thanks for your rebuttal. Many of my concerns have been addressed. I cannot see this paper 100% ready for being accepted but, I would not mind if it will be.
> >
> > I'm raising my score.
> >
> > Thanks

---

> > > ### Author Response · Authors · 2025-11-24
> > >
> > > We sincerely thank you for your continued engagement and for raising your score. We are greatly encouraged that our responses and revisions have satisfactorily addressed your concerns.
> > >
> > > We remain available for any further discussions should you have additional questions.

---

### Comment · Area_Chair_fiM3 · 2025-11-26

Dear Reviewers,

The authors have posted a detailed rebuttal. Please engage. This is a part of your reviewing responsibilities.

best,

AC

---

### Meta-Review · Area_Chair_jN6W · 2025-12-03

**Summary:**

The paper proposes a new framework for global counterfactual explanations in Heterogeneous Graph Neural Networks by perturbing meta-paths via a "path exclusion" algorithm. They shift the unit of explanation from nodes/edges to semantically meaningful meta-paths.   They conduct experiments with relevant baselines on fidelity, distance, and sparsity. Their initial experiments suggest improvements when using their framework over existing methods.

**Reviewer Concerns:**

Reviewers have raised significant concerns, which are highly valid.

Reviewer yj6y identified a critical flaw: the definitions and reporting of core evaluation metrics (Fidelity, Sparsity) were inconsistent with cited literature (inverted optimization direction) and values were reported without clarification as percentages, exceeding the theoretical [0,1] range.

The authors revised accordingly and added some clarifications. However, the descriptions and notations remain highly inconsistent. For example, in the revised version, Fidelity is defined as "the proportion of predictions that change under the counterfactual." However, as shown in the appendix, it appears to be the proportion of unchanged predictions. The usage of "Sparsity" is also confusing. Typically, it measures the percentage of kept edges, not perturbed edges. Additionally, the sparsity values in the rebuttal and the paper are inconsistent.

Presentation and Clarity: Multiple reviewers (yj6y, SWZT, hBoM) found the writing, table presentations (especially Table 1), and technical descriptions lacking. The paper was deemed difficult to follow, with important details relegated to the appendix. For example, Figure 4 is hard to interpret.

In Appendix E.3, a table or figure is necessary. In Appendix E.5, the complexity notation "O(log(dist))X" is non-standard—it is unclear how to interpret this when X represents big-O complexity (e.g., would O(N²) yield "O(log(dist))O(N²)" or "O(log(dist))N²"?).

**Reviewer Scores:**

yj6y: Explicitly stated they raised their score. Final: 6.
HWKH: Initially a 6.  I think it will be maintained.
SWZT: some concerns were addressed, 4->4/6.
hBoM: Initially a 4. Some concerns (more experiments, technical details) were addressed with new experiments and commitments. 4->4/6

---

### Decision · Program_Chairs · 2026-01-26

Reject